# How Do Green Investments, Foreign Direct Investment, and Renewable Energy Impact CO$_2$ Emissions? Measuring the Role of Education in E-7 Nations

**Pengtao Xu [1,2,*], Jianguang Zhang [2] and Usman Mehmood [3]**

1   School of Economics, Zhejiang University, Hangzhou 310000, China
2   Hangzhou Finance and Investment Group CO., Ltd., Hangzhou 310000, China; zhangjianguang@hzfi.cn
3   Department of Political Science, University of Management and Technology, Lahore 54792, Pakistan; usmanmehmood.umt@gmail.com
*   Correspondence: 0622731@zju.edu.cn

**Abstract:** The COP27 conference establishes fresh objectives for global economies to achieve the goals outlined in the Paris Agreement, which are centered on reducing carbon (CO$_2$) emissions and constraining the rise in global temperatures to 1.5 °C. In this background, this study looks at how education has affected CO$_2$ emissions, the economy, the use of renewable energy, green investments, and foreign direct investment in the E-7 countries from 2000 to 2021. Two unit root tests, CADF and CIPS, were used to gauge the data's stationarity. The long-run coefficients were identified using the momentum quantile regression approach. The empirical results show a cointegration of the variables. Long-term CO$_2$ emissions are influenced by a variety of factors, including foreign direct investment, economic growth, green investments, and education. The outcomes of reliable statistics provide support for the overall empirical study of groups and the economy. The results also suggest that there is a significant increase in education, leading to a reduction in CO$_2$ emissions across long time periods. Additionally, the E-7 countries should place a high priority on boosting the use of renewable energy and investing in the expansion of higher education for sustainable development. To mitigate the rise in carbon dioxide emissions (CO$_2$em), it is recommended that the governments of the E-7 nations take measures to promote the adoption of green investments. Governments must prioritize their efforts to ensure that green financing policies are able to complement environmental welfare policies and green growth policies.

**Keywords:** E-7 nations; green investments; renewable energy; education; FDI

## 1. Introduction

The issue of environmental degradation has emerged as a pervasive global concern [1,2]. Consequently, authors have expanded their investigations into environmental studies to mitigate the adverse repercussions and safeguard the integrity of the planet's ecosystem. In the context of national priorities, the fundamental goal of any nation is to enhance its economic progress (GDP) rate to foster social welfare. In this scenario, the occurrence of swift GDP can lead to the reduction of resources and the exacerbation of the environment [3]. Sustainable GDP in developing nations improves society. Trade, development, foreign direct investment (FDI), and natural resource (NAT) development have been used to achieve this goal. Production increases energy use and CO$_2$ emissions (CO$_2$em). GDP must not harm future generations [4,5].

It is evident that emerging economies such as the E-7 nations (Brazil, India, China, Indonesia, Mexico, Russia, and Turkey) still have a considerable distance to traverse to achieve sustainable development [6]. The rapid economic development of the E-7 countries has positioned them as significant contributors to global decision-making processes.

The countries mentioned in the study conducted by [7] represent a significant proportion, exceeding 42% of the global fossil fuel consumption, surpassing the consumption of the G7 countries. Furthermore, as stated in the "Global $CO_2$ Emissions Report 2019", developed economies experienced an average GDP rate of 1.7% in 2019. However, it is noteworthy that there was a notable decline of 3.2% in total energy-related $CO_2$ emissions during the same period. Nevertheless, it is worth noting that the electric power sector has played a significant role in the reduction of emissions, accounting for approximately 36% of energy-related emissions in developed economies. In contrast, in E7 countries, this sector is responsible for more than 47% of $CO_2$ emissions. Consequently, the E7 nations are assuming a progressively significant role in the global energy market and climate change, encompassing both carbon dioxide emissions and energy consumption. According to [8], the present predicament faced by E7 countries revolves around the identification of dependable and cost-effective energy alternatives to supplant fossil fuel-based energy sources, all the while mitigating the release of greenhouse gas emissions. Accordingly, emission-mitigation measures must be implemented globally considering the alarming rate of increase in global $CO_2$ emissions [9,10]. More crucially, it has been well postulated in the literature [11] that greening certain macroeconomic measures can help to decouple $CO_2$ emissions from economic growth. Purchasing environmentally friendly supplies is crucial in this context. Ref. [12] suggest that any investment associated with equity should be classified as a green investment (GF). The flow of capital is directed, and establishments are encouraged to actively invest in green projects, which are essential for environmental protection, resource conservation, and economic development [13,14].

Environmental degradation is largely attributed to GDP [15,16]. However, conventional wisdom holds that early GDP harms the environment but later restores it [17]. The impact of GDP on climate change has been found to be predominantly negative. However, there are several strategies that can be implemented to mitigate this impact [18–21]. The paramount consideration lies in the necessity for both individuals and society to collaboratively undertake concerted endeavors to alleviate these effects [22]. The significance of a well-educated society should be acknowledged considering its proactive endeavors to safeguard the environment [23]. Education plays a substantial role in fostering economic development through diverse channels. In broad terms, a society that possesses a high level of education is better equipped to confront a wide range of challenges [24]. Education is society's most prized tool for learning new skills and adapting to new technologies [25]. Several researchers examined education's impact on the environment. Ref. [26] found that education is the environmental connection threshold. Secondary education improves environmental quality empirically. Ref. [27] proposed an education level at which nations start to see a significant positive impact on $CO_2$em. Thus, education boosts GDP, technological innovation, and production energy source discovery.

The existing body of research examines the correlation between FDI and GDP, the association between renewable energy (RE) consumption and GDP, and the causal connection among GDP, RE consumption, green investments, and $CO_2$ emissions. Nevertheless, the existing body of research on the moderating role of higher education is relatively scarce. E7 countries exhibit a significant correlation with respect to energy demand, $CO_2$ emissions, and population levels in comparison to other countries globally. However, there is a scarcity of research studies addressing the current situation in these countries. At present, there exists a dearth of comprehensive empirical studies examining the effects of GDP, RE, FDI, and green investments on carbon emissions, while also considering the controlling influence on education within the E-7 countries. Hence, the present study aims to enhance the analysis by incorporating crucial factors pertaining to interconnectivity through the utilization of an E-7 panel dataset spanning the years 2000 to 2021. For analysis, this work uses the MMQR model. MMQR panel estimation examines the relationship concerning variables across multiple quantiles. The technique by Roger Koenker (2001) [28] is generally used to approximate the linkages between several factors at different quantiles. Quantile

regression is a statistical method that is resistant to the influence of outliers and generates effective estimates for datasets with heavy tails.

This research yields three primary contributions. Firstly, it employs robust estimators, which are analytically appropriate, in its analysis of panel data. Furthermore, the independent variable "portfolio" exhibits variation that differs from that observed in prior studies. Moreover, a clearly defined experimental model is utilized. This study utilizes up-to-date data to obtain empirical findings, with the aim of helping governments and policymakers in the formulation of effective public policies in E-7 countries. Additionally, this measure will contribute to the enhancement of GDP in these economies.

The structure of this work is as follows: after the introduction, the literature review is in Section 2, while the data and methodology are in Section 3. The results and discussion are in Section 4, and the conclusion is in Section 5.

## 2. Background Literature

### 2.1. $CO_2$em and Renewable Energy

Since RE technology is long-lasting and widely available, the authors of [29] recommended using it to reduce environmental degradation. Ref. [29] found that RE reduces environmental degradation and is sustainable. Countries need to increase RE consumption to reduce environmental damage. RE fuels industrialization, potentially improving the environment [30]. RE is more environmentally friendly than fossils [31]. The exponential growth in the utilization of RE sources plays a crucial role in mitigating the adverse environmental consequences [32]. Prior research has examined the relationship between RE consumption and its impact on $CO_2$ emissions, GDP, and environmental pollution [33].

The topic of RE has been extensively discussed and analyzed in various academic studies [34]. For example, Ref. [35] have demonstrated a significant positive relationship between carbon dioxide ($CO_2$) and RE.

In a similar vein, Ref. [36] conducted an empirical examination of the relationship between RE and $CO_2$em in China. The study utilized a dataset spanning the years 1980 to 2016, employing the Environmental Kuznets Curve (EKC) framework. The findings of the study indicate that the implementation of RE leads to a significant reduction in carbon dioxide emissions. Ref. [37] examined RE and $CO_2$em in the countries by using the Fourier autoregressive distributed lag (ARDL) cointegration, and found that RE lowers $CO_2$ in the BRICS economies. Ref. [38] used GS2SLS to examine RE and $CO_2$ emissions from 1995 to 2014, and found that RE predicts $CO_2$ emissions because RE causes them. Ref. [39] used a novel QQ approach to study $CO_2$ and RE in the US. They found that RE reduces $CO_2$ emissions. Finally, Ref. [40] found that RE reduces $CO_2$ emissions in Argentina while non-RE increases them.

While these research studies highlight the significant importance of RE for fostering GDP, several other research studies argue that the utilization of RE sources hinders GDP. Abbasi et al., 2020 and Baz et al., 2021 [41,42] conducted empirical investigations on various groups, including developed and developing economies, across different time periods. Their data analysis indicated that the slower GDP rate in these areas could be attributed to the utilization of RE sources. The relationship has generated conflicting arguments, leading to a renewed examination of this connection in emerging economies like BRICS.

### 2.2. $CO_2$em and Economic Progress

The increase in global carbon dioxide ($CO_2$) emissions is significantly influenced by recent economic development [43]. The E-7 countries are experiencing a significant transformation in their economic structure, resulting in a detrimental rise in $CO_2$em. Coal is the predominant source of energy in South Africa, thereby presenting a substantial risk to air quality. Similarly, Russia is heavily dependent on the steel, oil, and coal sectors, resulting in a significant escalation of $CO_2$em across the nation. Russia is positioned as the fifth highest emitter of greenhouse gases per capita globally.

Numerous contemporary studies have investigated the correlation between GDP and $CO_2$em. Hence, the empirical findings exhibit inconsistency when considering various factors such as the countries under investigation, the timeframe of the study, the approaches employed, and the techniques utilized. Ref. [44] utilized the wavelet method to investigate the relationship between $CO_2$em and economic progress. A positive linkage was observed between the levels of $CO_2$ emissions in Indonesia and the country's economic development. Similarly, Ref. [45] gathered data spanning from 1971 to 2016 to analyze the influence of $CO_2$em on the GDP of China. The researchers discovered that there is a positive correlation between GDP and $CO_2$em across quantiles (0.1–0.95) when employing the QQR technique. This suggests that the rapid GDP in China has detrimental effects on the environment. In a similar vein, Ref. [46] conducted a study utilizing data spanning from 1971 to 2017. They employed a unique two-gap methodology to examine the relationship between GDP and $CO_2$ emissions in Turkey. The researchers discovered that the pro-growth policy implemented in Turkey is responsible for the environmental degradation observed within the nation.

Furthermore, Ref. [47] utilized frequency domain causality and innovative two-gap methodologies to examine the relationship between $CO_2$em and GDP in Mexico. The findings demonstrate a positive correlation between the growth of Mexico's economy and the escalation of environmental degradation within the country. In contrast, several research findings indicate a negative correlation between GDP and $CO_2$em. Ref. [48] utilized a novel quantile-on-quantile methodology to investigate the association between GDP and $CO_2$ emissions in Sweden, using a dataset spanning the years 1965 to 2019. The researchers discovered that the economic expansion observed in Sweden is associated with a reduction in $CO_2$em, suggesting the presence of sustainable growth within the nation.

*2.3. $CO_2$em and Green Investments*

The term "green investments (GF)" refers to services and goods that have an impact on the environment. According to [49], the implementation of GF has been found to reduce energy limitations, leading to favorable outcomes in terms of $CO_2$em and economic progress. Countries across the globe have commenced allocating financial resources towards diverse green initiatives to facilitate the realization of green GDP. The implementation of these initiatives is closely linked to the preservation of the environment and the achievement of desired outcomes [50]. Ref. [51] proposes the implementation of various advanced technologies to ensure sustainable development. The implementation of green financing for diverse projects can facilitate stakeholders in allocating their research and development funds towards matters pertaining to environmental sustainability [52]. Additionally, the implementation of green financing can help mitigate the financial constraints, enabling stakeholders to make investments in sustainable practices.

GF facilities can help private firms meet these goals [53]. Thus, green financing may make balancing economic development and environmental protection harder. Although important, the relationship between GF and environmental degradation is unexplored. Stakeholders (regulators/governments/organizations) who profit from environmental policies may strategically participate in green financing. Stakeholders must understand GF's benefits. Green financing may depend on environmental improvement [54]. Ref. [55] predicted that GF could reduce fossil fuel consumption by 2.5% by 2030. They also found that RE will generate 46% of global electricity. The authors of [56] used a fixed-effects model to study the association between digital finance and innovation in urban areas utilizing city-level panel data collected in 268 Chinese cities from 2011 to 2019. They found that urban innovation can be effectively increased by digital finance, yet there are notable variations in several areas. Moreover, towns with varying degrees of commercial attractiveness exhibit varied effects of financial technology on urban innovation. The promotional effect of digital money on urban innovation is moderated by the availability of traditional finance. Moreover, in [57] the authors developed a comprehensive measurement for assessing the level of corporate environmental responsibility (CER) engagement. This

measurement is utilized to investigate the correlation between CER engagement and firm value. Additionally, the researchers aimed to explore the potential mediating role of corporate innovation in this relationship. The study was conducted using a sample of 496 companies listed on China's A-share market, spanning the period from 2008 to 2016. The findings indicate that the implementation of environmental rules by enterprises initially leads to a decrease in firm value. However, at a certain threshold, the adoption of these policies begins to have a beneficial impact on firm value. Furthermore, it is worth noting that corporate innovation serves as a mediator in the correlation between corporate environmental responsibility (CER) and business value. The implementation of corporate innovation initiatives has been found to have a greater positive impact on the overall value of organizations that possess a Corporate Entrepreneurship Readiness (CER) compared to firms that lack such readiness.

GF includes any production-efficiency-boosting expenditure. Unexpectedly little research has been done on the association between GF and the environment. The impact of GF on $CO_2$ emissions and RE, however, is rarely highlighted in research. For instance, GF significantly affects sustainable development [58]. According to [59], $CO_2$ emissions are decreased by private investment in environmentally friendly projects. Additionally, such investment aids developing nations in embracing a green mindset [60]. According to [61], the contemporary media landscape has the potential to incentivize environmentally detrimental firms to address the expectations of their stakeholders and make substantial advancements in their adoption of sustainable technology. The utilization of diverse environmental regulatory instruments, such as pollution charges and environmental protection subsidies, can have a collective positive impact on the advancement of green technological innovation within corporations. This is achieved through the combined influence of pushback and compensation effects. Furthermore, the implementation of effective environmental regulation tools can strengthen the relationship between the government and enterprises, and enhance the preparedness of heavily polluting enterprises in terms of resources and dynamic capabilities to effectively address public opinion crises. Consequently, this can serve as a moderating factor in the promotion of new media environments, ultimately fostering corporate green technological innovation. Additionally, the impact of the digital economy on national-level industrial eco-efficiency is predominantly positive, with a decrease in marginal returns. The impact of the internet economy on industrial eco-efficiency exhibits considerable variation across different regions. The impact of the digital economy on corporate eco-efficiency is found to be notably good in eastern regions, but it is observed to be detrimental in western regions. The impact of the digital economy on corporate eco-efficiency in China is shown to be negligible, suggesting the presence of digital isolation [62]. The concept of green finance encompasses various interconnected principles. This work provides a concise description of the global events that have contributed to the evolution of green finance, the common forms and instruments utilized in this field, the regulatory framework and issuance process associated with these instruments, and the diverse international agencies and organizations involved in the development and implementation of green finance schemes for specific beneficiary projects [63].

*2.4. $CO_2$em and FDI*

In a recent study by [64], it has been proposed that world economics aims to enhance its economic prospects through the adoption of a multidimensional approach to globalization. Financial globalization and GDP are closely associated with various forms of globalization.

Hence, it can be argued that financial globalization is accountable for the concurrent escalation of GDP and environmental degradation, as evidenced by the increased influx of FDI [65]. According to [66], financial globalization emerges as a prominent factor in the examination of environmental quality. Nevertheless, it is worth noting that financial investments can serve as a catalyst for industrialization, which in turn may result in adverse environmental consequences such as increased degradation and pollution [67].

In [68], among other recent studies, the authors contend that FDI promotes GDP. Additionally, it generates employment opportunities and makes international technology transfers easier. Moreover, international trade activities raise $CO_2$ emissions globally despite environmental agreements [69]. Furthermore, many nations that support financial globalization strategies to achieve globalization-driven GDP view the global FDI data as a significant predictor of $CO_2$ emissions). However, one of the major worries about FDI investments is the potential harm to environmental quality [70]. Since the negative impacts of FDI investments on environmental degradation are disregarded, the economic prospects associated with FDI investments are not maintainable [71]. FDI inflow, mostly via technology routes, positively impacts the climate [72]. Ref. [72] found that FDI boosts GDP and reduces $CO_2$em. Current research shows that GDP and energy use degrade the environment, but RE and trade via FDI may mitigate these effects.

More usage of RE and FDI information reduce environmental pollution in the American and Asian regions. Ref. [73] conducted an empirical investigation of the association between FDI and environment in China. They found that GDP brought on by FDI enhances the environment by reducing fossil fuel consumption and increasing efficiency. Therefore, one can discuss the economic impact of FDI information on the environment.

*2.5. $CO_2$em and Education*

Because education has so many implications for the advancement of knowledge and technology, many people think that the growth of a nation is inversely correlated with the caliber of its education arrangement. Previous findings from research studies already conducted represent two distinct areas of investigation.

While the later feature of the research work bases the positive effect of education in lowering $CO_2$em, the first aspect emphasizes performances that create high $CO_2$em. The first group of proponents argues that because higher education institutions offer a wide range of academic activities, education is to blame for rising $CO_2$em.

Ref. [74] conducted a recent study in Spain wherein they developed a metric to quantify emission activities within a university campus. The study conducted by the researchers revealed that the average $CO_2$em produced by individuals attending educational institutions in the United States amounted to approximately 41 metric tons per person, specifically attributed to transportation activities on campus. Ref. [75] conducted a study in Spain utilizing a dataset spanning from 2011 to 2014. Their findings revealed a consistent trend in $CO_2$ emissions, with approximately 0.55t $CO_2$ attributed to various on-campus activities within higher education institutions. Ref. [76] argue that the implementation of online education is imperative to mitigate $CO_2$ emissions within the higher education sector, thereby reducing the global and regional carbon footprint. This phenomenon occurs because of the significant carbon dioxide emissions stemming from the frequent transportation undertaken by employees and students within the Dutch institutions under examination. The average $CO_2$em per person resulting from housing, mobility, air transportation, food, and consumption were approximately 10.9 metric tons of $CO_2$. Nevertheless, the heat emissions produced by students were minimal.

The 2nd part of the investigation confirms education's positive impact on $CO_2$ emissions using various methods tailored to individual nations. Ref. [77] estimated China's education's $CO_2$ emissions percentage. Higher education affects national and regional environments along with other demographic structures. In another Chinese region, higher education was negatively correlated with $CO_2$ emissions. The ratio was calculated from the 6+ student percentage. Ref. [78] examined education, environmental pollution, and poverty in 22 developing countries using panel data. Their results show that education reduced the negative impacts of the environment. Ref. [79] found that a year of environmental programs reduced $CO_2$em by 2.86 tons. They stressed environmental efforts.

After discussing the literature review, it is evident that very few studies investigated the impacts of education, renewable energy, and green investment on $CO_2$ emissions in E-7 countries. Most of studies have considered human capital and very few of them consider

education expenditures for other regions. Therefore, to contribute to the literature and for updated evidence, this work adds education as a moderator in the linkages of green investment, renewable energy, FDI, GDP, and $CO_2$ emissions in E-7 nations.

### 2.6. Theoretical Framework

This study investigates a variety of factors, including economic growth, educational spending, energy from renewable sources, green investments, and foreign direct investment, that affect $CO_2$ emissions. $CO_2$ emissions are typically positively impacted by economic growth. High-growth economies frequently see rising income levels among their citizens, which encourages more consumption and ultimately raises $CO_2$ emissions. Economies are expanding their economic activity by utilizing renewable and non-renewable energy. These economic activities are making different countries increase FDI. These activities are contaminating the air quality [80]. Environmental awareness is among the fundamental factors that can speed up the carbon neutrality process. Through education, environmental awareness can be spread and pollution can be controlled [81]. For instance, Ref. [82] looked at how education significantly lowers $CO_2$ emissions in society. Furthermore, a proposed research model [83] revealed a significant relationship between GF and $CO_2$ emissions. According to a research model [24], RE lowers $CO_2$ emissions at both the governmental and personal levels. The promotion and advancement of GDP in emerging economies, such as the E-7 countries, is greatly aided by FDI.

The research model shows that, among other factors, GDP encourages business expansion and FDI in an economy. According to [84], FDI helps the industrial sector expand and opens job opportunities that increase demand for skilled workers. The government places significant emphasis on allocating resources towards the advancement of education, spanning from primary to tertiary institutions, as well as research facilities and vocational training centers. This strategic investment aims to address the increasing demand for proficient workers in both domestic and international corporations [85]. Broadly speaking, the phenomenon of large-scale expansion elucidates certain energy-economic activities that incur considerably higher costs. This scenario pertains to domains characterized by a substantial increase in both the expansion of educational institutions and the enrollment of students.

For instance, E-7 member nations frequently encounter such experiences. Infrastructure development includes these endeavors as well as the formation and administration of educational services, dorms for students and teachers, and office buildings. Mobility activities can also include staff and student travel to and from classes on or off campus, as well as trips for food, shopping, and medical attention. As was already mentioned, the consumption of non-RE during campus operations significantly raises the carbon footprint.

The acquisition of advanced knowledge, skills, and technologies through education plays a crucial role in facilitating the process of economic development. This, in turn, creates an enabling environment for the adoption of sustainable practices, including the increased utilization of renewable energy sources and the implementation of green financing mechanisms. The expansion of educational standards typically necessitates the allocation of financial resources and the provision of incentives to support innovative and competitive research endeavors aimed at developing alternative energy models, advanced technologies, and patents. The conceptual framework of this work is shown in Figure 1.

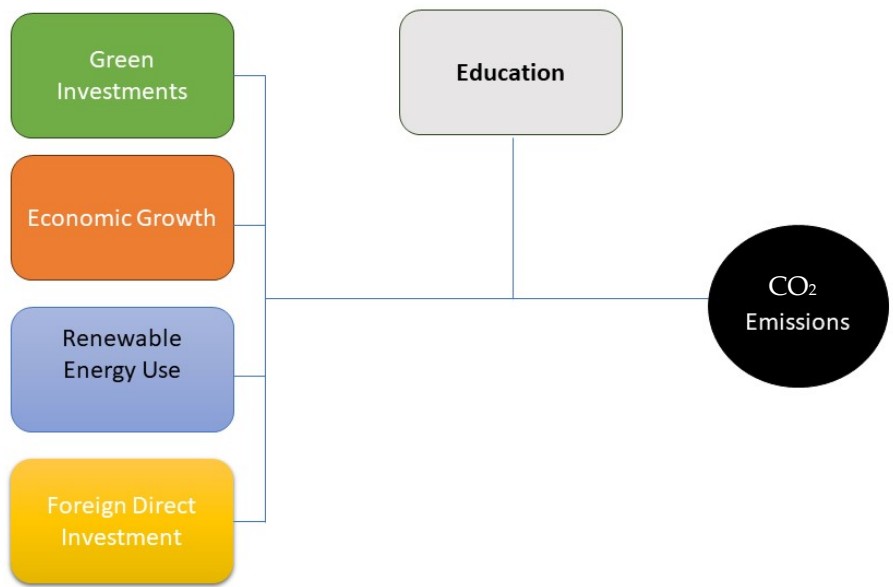

**Figure 1.** Graphical form of the study.

### 3. Data and Methodology

Incorporating the moderating effect of education in the E-7 countries, the underlying research work examines the relationship between economic growth, renewable energy consumption, green investments, foreign direct investment, and $CO_2$em. The E-7 sample shows significant $CO_2$em, and some of these nations are major carbon emitters, which is why the E-7 economies were chosen for investigation. Because there were available data from 2000 to 2021, we used balanced panel data. Economic growth (estimated as GDP per capita), renewable energy consumption (renewable energy consumption as % of total final energy consumption), green investments (a composite index via principal component analysis using the variables of number of patents, energy efficiency, and renewable energy), and foreign direct investment (FDI) are all listed as independent variables. This work follows the work of [86] to construct an index of GF. As a dependent variable, $CO_2$ emissions are included and are expressed as tons of $CO_2$ per capita.

In the analysis, the role of education is used as a moderating variable. Researchers have used a variety of proxies to estimate economic growth based on prior research. Several well-known studies, including [87,88], contend that while many factors influence a nation's economic growth, GDP per capita can be regarded as a highly reliable indicator. Studies such as [89–91] provided guidance for the choice of additional variables and the use of proxy measures. The information for the proposed variables was gathered from several reliable sources. For example, the World Bank's database was chosen to gather information on FDI information, government spending on education, and GDP per capita. Since the dataset on education contains missing observations, we must first identify these values before we can use a statistical model. These missing values in the dataset are located using the confidence interval method. This method examines each value as it moves from the first to the last element in a sequential order and identifies the observations that are missing. Then, using the mean and standard error of the available observations, it determines the confidence interval using the confidence interval technique. The median of the confidence interval is then used to replace the missing values. The data, their units and sources are mentioned in Table 1.

**Table 1.** Variables and their sources.

| Parameters | Symbol | Unit | Source |
|---|---|---|---|
| **CO$_2$ emissions** | CO$_2$ | CO$_2$ emissions metric tons per capita | WDI |
| **Economic progress** | GDP | GDP per capita constant (2015) | WDI |
| **Renewable energy** | RE | Renewable energy % of total energy use | WDI |
| **Green investments** | GF | A composite index via principal component analysis using the variables of number of patents, energy efficiency, and renewable energy | WDI |
| **Foreign direct investment** | FDI | Foreign direct investment inflows % of GDP | WDI |
| **Education** | EDU | Education expenditures by government (% of government expenditures) | WDI |
| **Environmental technology** | ENT | Patents on environmental technology | [92] |

*Methodology*

The interdependence among cross-sectional units has garnered significant attention in contemporary literature, resulting in numerous empirical investigations. The reason for this is that residuals exhibit a lack of independence within the actual context. Therefore, interdependence is inherent. The regional economies that are in the E-7 exhibit cross-border interconnections across various domains, including political, economic, social, environmental, and financial spheres. The correlation implies that any empirical inquiry conducted on these economies must consider the presence of cross-sectional dependence (CD). Equation (1) is proposed for the purpose of investigating the CD in the data.

$$\text{CD}_{TM} = \left[ \frac{TN(N-1)}{2} \right]^{\frac{1}{2}} \bar{\rho}_N \tag{1}$$

The correlation between the parameters is represented by $\rho_N$ over the time interval T. Furthermore, an inquiry into the heterogeneity present in the slope coefficient has been carried out using the Pesaran and Yamagata (2008) test, which is a revised variant of Swamy's (1970) test [93]. Equation (2) has been proposed in this context.

$$\widetilde{\Delta}_{ASH} = (N)^{\frac{1}{2}} \left( \frac{2k(T-k-1)}{T+1} \right)^{-\frac{1}{2}} \left( \frac{1}{N} \widetilde{S} - 2k \right) \tag{2}$$

In the context being discussed, *N* is used to represent the cross-sectional units, while K is utilized to denote the explanatory variables. Upon conducting an examination of the heterogeneity present in the slope coefficients, analysis has been carried out to determine the order of integration between the variables using both the cross-sectionally augmented IPS (CIPS) test and the cointegration augmented Dickey–Fuller (CADF) test. Nonetheless, it is imperative for researchers to give considerable attention to the matter of outcomes during regression estimation.

Hence, the present research has utilized the stationarity test, wherein the cointegration augmented Dickey–Fuller (CADF) test has been elucidated in Equation (3) of this study.

$$\Delta Y_{i,t} = \gamma_i + \gamma_i Y_{i,t-1} + \gamma_i \overline{X}_{t-1} + \sum_{l=0}^{p} \gamma_{il} \Delta \overline{Y_{t-l}} + \sum_{l=1}^{p} \gamma_{il} \Delta Y_{i,t-l} + \varepsilon_{it} \tag{3}$$

The lagged parameter denoted as $Y_{t-1}$ and the initial difference of $Y_{t-1}$ is represented by $\Delta Y_{t-1}$. The computation of CIPS statistics involves the determination of the mean of CADF, which is elucidated in the following equation:

$$\widehat{CIPS} = \frac{1}{N} \sum_{i=1}^{n} CADF_i \tag{4}$$

Subsequently, an examination of the cointegrated relationship among the specified factors, namely, renewable energy, GDP, education, and green investments was carried out using Westerlund's cointegration test. After confirming the cointegration, this work moves forward to apply a novel nonlinear econometric method, namely, the Method of Moments Quantile Regression (MMQR) by [94]. Traditional linear econometric methods have only focused on modelling the mean of panel data, rather than the conditional distribution [95]. In contrast, MMQR panel estimation examines the relationship concerning variables across multiple quantiles. The technique by Roger Koenker (2005) [96] is generally used to approximate the linkages between several factors at different quantiles. Quantile regression is a statistical method that is resistant to the influence of outliers and generates effective estimates for datasets with heavy tails. According to [97], the method maintains consistency even when multicollinearity is present. However, it should be noted that the quantile regression model exhibits a limitation in its ability to ensure noncrossing outcomes for a multitude of percentiles, which may lead to an inaccurate representation of the response distribution. Considering the context, it is recommended to utilize MMQR owing to a multitude of factors. The model yields consistent results even in the presence of unobserved endogeneity and heterogeneity across the cross-sections. The MMQR methodology allows for the conditional and heterogeneous influence of ecological footprint determinants to impact the distribution's quantiles. This approach proves advantageous in cases where explanatory variables exhibit high correlation and endogenous behavior.

This method works for high-kurtosis nonlinear datasets. It captures data dynamics endogenously, making it better than other nonlinear modelling methods [98]. Since parameters depend on response variable location, MMQR allows for asymmetric variable location. Partial-parametric modelling structures like MMQR are ideal for dealing with asymmetry, heterogeneity, and endogeneity producing estimates across numerous quantiles [99]. The amended location-scale definition for conditional quantiles Q(X) is

$$y_{it} = \beta_i + X_{it}\alpha + (\delta_i + U_{it}\gamma)c_{it}$$

The probability (p) can be expressed as $P(\delta_i + U_{it}\gamma > 0) = 1$, where $(\beta, \alpha, \delta, \gamma)$ are the assessed factors $(\beta_i, \delta_i)$, $i = 1, \ldots N$, that confirm the fixed effects of individual $i$. Here, $U$ represents a chosen j-vector element of $X$ that accounts for difference transformation in the equation, denoted as Ul = Ul(X), l = 1, ……..j. Furthermore, it can be observed that $X_{it}$ denotes an equivalent distribution for a given individual at a distinct point in time (T). Assuming identical distribution at an individual level (i) for time (T), Zit is impertinent to $X_{it}$, Machado and Silva (2019) [94] instants conditions. The model can be expressed in its quantile in the following manner:

$$Q\tau\left(\frac{\tau}{X}\right) = (\beta_i + \delta i_p(\tau)) + X_{it}\alpha + U_{it\gamma p}$$

The variables EDU, FDI, GDP, GF, and RE are represented as $X_{it}$, while $Q\tau(\tau/X)$ denotes the dependent factor $Y_{it}$, which is the $CO_2$, conditioned on fundamental quantiles

and placed as the independent factor. The notation $\beta_i(\tau) = \beta_i + \delta_{i\mathrm{p}}(\tau)$ is used to denote the quantile $\tau$i for an individual. The parameter attains a fixed state and exhibits significant heterogeneous effects, thereby enabling the quantile model $\tau$th to manifest through $q(\tau)$ derived from the linearity problem.

$$\min_q \sum_i \sum_t \pi_\tau (W_{it} - (\vartheta_{it} + Z_{it}\theta)q)$$

The check function, denoted by the equation above, is

$$\pi_\tau(A) = (\tau - 1)AI\{A \le 0\} + TAI\{A > 0\}$$

## 4. Results and Discussion

This work estimates the panel data of E-7 nations. The descriptive statistics are presented in Table 2.

**Table 2.** Descriptive statistics.

|  | $CO_2$ | EDU | FDI | GDP | GF | RE |
|---|---|---|---|---|---|---|
| Mean | 4.625312 | 7.244516 | 1.993958 | 5301.884 | 8.070167 | 18.46964 |
| Median | 3.985111 | 4.773390 | 1.929270 | 5471.307 | 7.890720 | 13.62875 |
| Maximum | 11.88496 | 16.73051 | 4.554254 | 11938.78 | 13.63472 | 47.11000 |
| Minimum | 0.883747 | 2.390000 | −2.757440 | −7.138251 | 4.013677 | 3.180000 |
| Std. Dev. | 3.169755 | 4.476989 | 1.128676 | 3908.736 | 2.227632 | 12.58527 |
| Skewness | 1.112224 | 0.645260 | −0.338514 | 0.020365 | 0.661060 | 0.854173 |
| Kurtosis | 3.203081 | 1.858071 | 4.796409 | 1.451163 | 3.117271 | 2.550081 |
| Jarque–Bera | 32.01539 | 19.05394 | 23.64832 | 15.40356 | 11.30459 | 20.02562 |
| Probability | 0.000000 | 0.000073 | 0.000007 | 0.000452 | 0.003509 | 0.000045 |

Table 2 shows that GDP has the highest mean and FDI has the lowest mean. $CO_2$em has a maximum value of 11.89 and a minimum value of 0.89. RE has a maximum value of 47.11 and a minimum value of 3.18. The graphical form of the descriptive statistics is in Figure 2.

The next step is to find out the cross-sectional dependence (CD) in the panel data of the E-7 nations. The CD test provides the data description and ensures the applicability of subsequent tests. Table 3 provides the results, and it shows that all the variables are having CD at the 1% level.

**Table 3.** CD test.

| Variable | Test Statistics |
|---|---|
| $CO_2$ | 10.56 [a] |
| GDP | 13.96 [a] |
| RE | 14.66 [a] |
| GF | 15.54 [a] |
| FDI | 5.61 [a] |
| EDU | 10.20 [a] |

Note: [a] explains the level of significance at 1%.

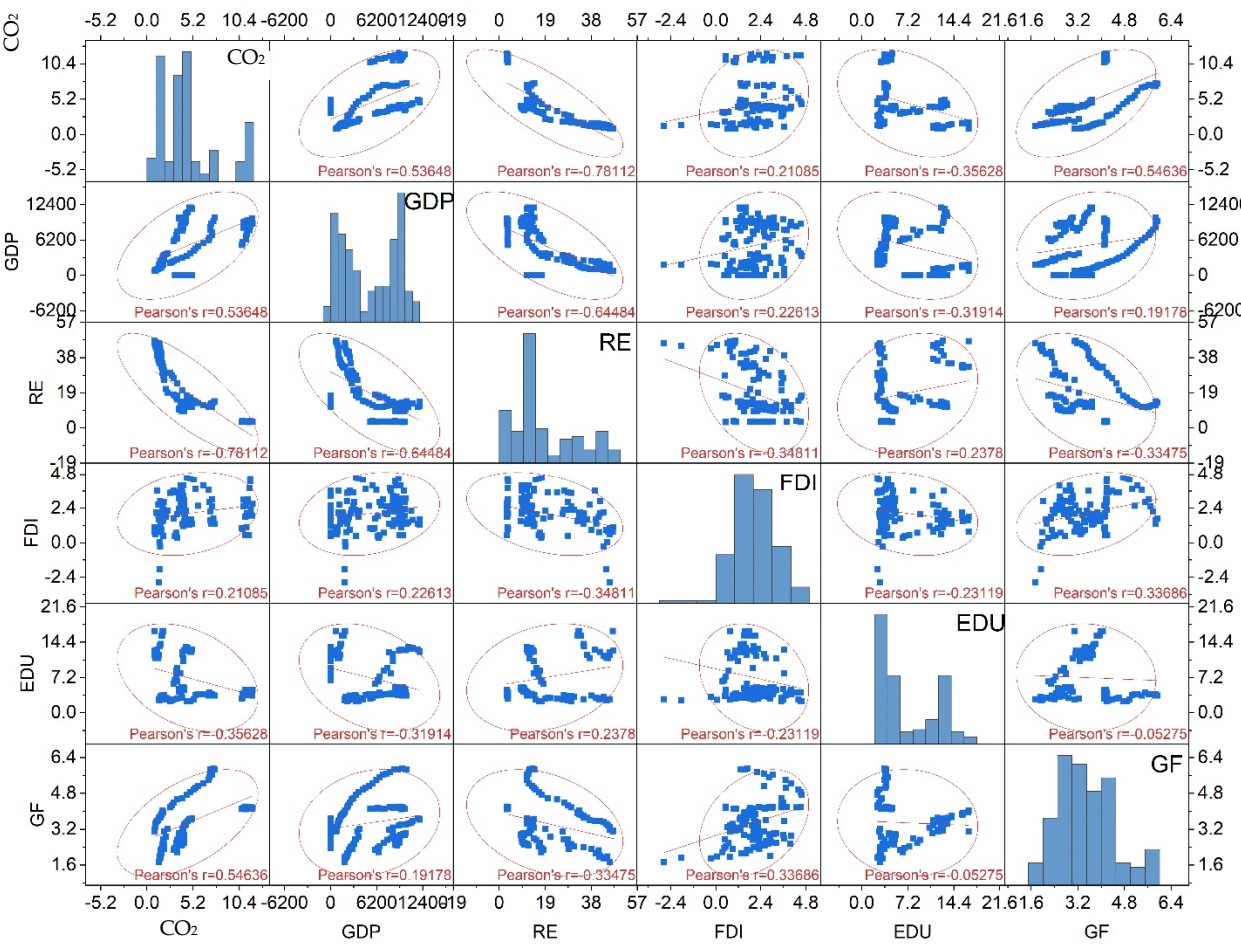

**Figure 2.** Scatter Plot.

The next step is to know the order of integration of the variables. Some variables may be integrated at level, and some may be at first difference. For this purpose, this work applies the CIPS and CADF unit tests. These tests are better than the first unit root tests of [100]. Table 4 shows the findings of both unit root tests.

**Table 4.** Unit root tests.

| Variable | CADF Test | | CIPS | |
|---|---|---|---|---|
| | At Level | 1st Difference | At Level | 1st Difference |
| $lnCO_2$ | −1.858 | −2.938 [a] | −1.994 | −3.863 [a] |
| $lnGDP_t$ | −1.224 | −3.493 [a] | −2.505 | −3.493 [a] |
| $lnRE_t$ | −2.709 | −3.176 [a] | −2.543 | −4.083 [a] |
| $lnGF_t$ | −1.519 | −3.569 [a] | −2.357 | −5.533 [a] |
| $lnFDI_t$ | −3.515 | −5.070 [a] | −3.786 | −5.554 [a] |
| $lnEDU_t$ | −2.943 | −4.259 [a] | −3.241 | −5.430 [a] |

Note: [a] explains the level of significance at 1%.

Table 4 shows that all the variables (CO$_2$, GDP, RE, GF, FDI, and EDU) are cointegrated at first difference. The next step is to know the slope property of the panel data. Table 5 shows that the panel data are heterogeneous.

**Table 5.** Slope test.

|  | Value | *p* Value |
|---|---|---|
| **Delta** | 2.508 [b] | 0.012 |
| **adj** | 3.238 [a] | 0.001 |

Note: [a] and [b] explain the level of significance at 1% and 5%.

Before the long run analysis, it is essential to conduct a cointegration test. For this purpose, this work uses Westerlund's test [101]. This test provides robust results while considering the CD in the panel data.

Table 6 shows that the panel data are cointegrated because group and panel values are significant at 1% level. The coefficient values in the panel data further make it possible to know the long-run coefficient values. For the long-run analysis, this work applies the MMQR approach. This method provides the impacts of independent variables on the dependent factor along different quantiles. Next the MMQR results are in Table 7.

**Table 6.** Westerlund test.

| Stat | Value | Z Value | *p* Value |
|---|---|---|---|
| $G_t$ | −4.671 [a] | −5.622 | 0.000 |
| $G_a$ | −5.104 | 3.168 | 0.999 |
| $P_t$ | −13.618 [a] | −6.790 | 0.000 |
| $P_a$ | −5.659 | 1.796 | 0.964 |

Note: [a] explains the level of significance at 1%.

**Table 7.** MMQR results.

| | | | Quantiles | | | |
|---|---|---|---|---|---|---|
| **Variable** | **Location** | **Scale** | **0.25** | **0.50** | **0.75** | **0.90** |
| **GDP** | 0.070 *** | 0.026 *** | 0.013 *** | 0.015 ** | 0.034 * | 0.054 * |
| **RE** | −0.165 *** | 0.092 * | −0.158 *** | −0.161 *** | −0.181 *** | −0.181 *** |
| **GF** | −1.025 *** | −0.235 * | −1.069 *** | −1.143 *** | −1.484 *** | −0.625 *** |
| **FDI** | −0.559 * | −0.034 ** | −0.532 * | −0.543 * | −0.593 ** | −0.617 * |
| **EDU** | −0.151 *** | −0.096 *** | −0.076 *** | −0.106 *** | −0.246 *** | −0.315 ** |

Note. ***, **, and * indicate the level of significance at 1%, 5%, and 10%, respectively.

At all quantiles of $CO_2$ emissions, the estimated coefficient of GDP exhibits a significant positive relationship. The observed trend indicates a positive correlation between the impact of GDP and $CO_2$em, whereby a higher quantile level corresponds to a higher impact of GDP. Specifically, when $CO_2$em is situated at a higher quantile level, the effect of GDP is also observed to be higher. This finding aligns with the research conducted by [67,102–104]. The subtext of this statement is that the E-7 nations have experienced positive effects on their economy's core sectors, including farming, manufacturing, and transportation. Another potential factor to consider is that GDP may stimulate economic activity through the promotion of investment, purchasing, consumption, and energy use, thus leading to a rise in pollution levels [105].

The impact of RE is negative at all quantiles of $CO_2$em. This impact is lowering from the 25th to the 90th quantiles. RE is mostly generated from wind, hydro, and solar energy. These energies do not consume fossils and, therefore, do not harm the climate. These results are in line with the findings of [106].

As far as the impact of GF on $CO_2$em is concerned, our analysis shows that GF is environmentally friendly. This is not an alarming situation for E7 countries because the GF

has been able to lower environmental pollution, and it is not a contributing factor towards $CO_2$em. This work constructed the GF as an index composed of the three factors of energy efficiency, RE, and number of patents in E-7 nations. This implies that the allocation of resources towards environmentally friendly initiatives has played a role in reducing $CO_2$ emissions, thereby enhancing the overall climate conditions inside the nation.

The result also suggests that the E-7's GF policies are aligned with its objective of transitioning towards a low-carbon economy. An alternative hypothesis posits that the energy expenses incurred by industrial and other enterprises in the E-7 countries could be substantial. According to the research conducted by [107], companies exhibit reluctance towards adopting green technologies, such as green energy, when the energy costs in a particular country are relatively cheap. Entities that incur higher energy costs place a greater emphasis on considering the impact of green initiatives in their decision-making processes, in contrast to entities with lower energy expenditures. The research findings support the legitimacy theory, which argues that corporations should adhere to policies, regulations, and conventions that contribute to environmental sustainability, as evidenced by the negative correlation between GF and $CO_2$ emissions. Based on this disclosure, it may be inferred that businesses in the E-7 countries adhered to this principle by allocating resources towards the adoption of ecologically sustainable energy sources, machinery, and technology, among other relevant measures. The findings of [108–110] support the conclusions of this study. However, the findings of [111,112] differ from those mentioned above.

FDI is showing negative signs in all quantiles. This impact is continuously increasing. This means that FDI is suitable to lower the $CO_2$em in E-7 countries. FDI enables new opportunities for the other developed nations to invest in host countries. It further makes it possible to import efficient technologies from other developed nations into the host countries. These technologies contribute to the lowering of the environmental pollution. The role of education is also friendly in all quantiles. This means that education expenditures are lowering $CO_2$em in all quantiles, but this impact is lowering while moving to higher quantiles. Education expenditures create opportunities to educate people and to spread environmental awareness. This awareness further encourages citizens to adopt sustainable ways of life. This result is in line with the findings of [49].

*Robustness Check*

To check the validity of the MMQR results, this work applies the CCEMG methodology and takes environmental technologies (ENT) as a proxy of green investment. ENT are mainly crafted to deal with environmental pollution and are an authentic variable to measure green investments [113]. These data have been obtained from OECD. This method is efficient in providing robust results while incorporating the CD in the panel data. Table 8 shows that RE, GDP, FDI, and EDU impact negatively. This means that these factors are environmentally friendly in the E-7 nations. Green investments are increasing $CO_2$em. These results are in line with the MMQR results.

**Table 8.** CCEMG.

| Variable | Coefficient | Z-Value | *p*-Value |
|:---:|:---:|:---:|:---:|
| GDP | 0.155 * | 1.81 | 0.071 |
| RE | −0.140 *** | −2.77 | 0.006 |
| GF | −1.838 *** | −2.69 | 0.007 |
| FDI | −0.066 ** | −1.98 | 0.048 |
| EDU | −0.037 * | −1.61 | 0.091 |

Note. ***, **, and * indicate the significance of level at 1%, 5%, and 10%, respectively.

## 5. Conclusions and Policy Suggestions

This research investigates the relationship between carbon dioxide ($CO_2$) emissions, economic growth, renewable energy consumption (RE), green investments (GF), and foreign direct investment (FDI) in the context of higher education in the E-7 countries from 2000 to 2021. The stationarity of the data was assessed through the utilization of three distinct unit root tests. The results of the cointegration analysis indicate the existence of a cointegrating relationship among the variables. The MMQR model was utilized to examine the long-term dynamic relationships among the variables. The findings of the MMQR study indicate that there exists a negative relationship between economic growth, renewable energy (RE), foreign direct investment (FDI), and education (EDU) on the reduction of carbon dioxide ($CO_2$) emissions over an extended period. However, it is worth noting that green investments have the potential to contribute to an increase in $CO_2$ emissions. The findings of this study provide several significant recommendations for policymakers and governments to address the reduction of $CO_2$ emissions while simultaneously promoting sustainable economic growth. Initially, it is imperative for the E-7 economies to allocate a significant proportion of their financial resources towards the advancement of education. There is a need to expand vocational institutions and other educational facilities to accommodate the growing population. In addition to this, it is imperative for governments to give precedence to the establishment of research and development institutions. Through this approach, a multitude of individuals possessing exceptional qualifications will ultimately devise a resolution aimed at mitigating carbon dioxide emissions, facilitated by advancements in technology.

It is important to encourage the use of renewable energy sources like hydrogen, biofuel, biomass, solar, wind, and others as a means of reducing or eliminating our excessive reliance on fossil fuels for economic purposes. Economic incentives that support green services can be encouraged, such as tax waivers or discounted tax rates on goods and services. To secure a sufficient transition to 100% energy from renewable sources, as envisaged by the bulk of recent environmental treaties, the governments should specifically support renewable energy.

Similarly, encouraging investment in innovations, research, and development helps sustain the carbon-mitigating roles of green technology. The governments ought to encourage financial institutions that finance green projects and permit the private sector to participate more in these initiatives. It is difficult to overstate the importance of education in endogenizing advances in technology. Therefore, the governments of the different E7 economies ought to focus more on the education system, especially by reforming and creating curricula that foster the development of creative ideas and skilled labor.

Furthermore, collaboration between the government and policymakers should be undertaken to enact diverse taxation policies with the aim of regulating $CO_2$ emissions within the environment. The achievement of environmental sustainability can be attained through the expansion of renewable energy sources, as opposed to non-renewable sources, in both product manufacturing and power generation [114]. E-7 countries should consider implementing policies centered around a system of rewards and penalties to address the issue of environmental degradation. It is imperative for the governments of these nations to acknowledge and incentivize businesses and industries that comply with governmental regulations and employ environmentally sustainable energy sources to meet their production requirements. To mitigate the rise in carbon dioxide emissions, it is recommended that the governments of the E-7 nations take measures to promote the adoption of green investments. Governments must prioritize their efforts to ensure that green financing policies are able to complement environmental welfare policies and green growth policies. Finally, it is imperative for the E-7 nations to optimize the utilization of FDI within their respective economies. To ensure long-term economic sustainability, it is imperative for the governments of the E-7 countries to implement stringent regulatory measures to monitor and control the activities of multinational corporations within their jurisdictions.

This approach aims to mitigate environmental degradation and foster the advancement of sustainable development.

Alongside these contributions, this work has some limitations that should be addressed by upcoming research. This work adopts an index of renewable energy, patents, and energy efficiency. Future research can incorporate other proxies of green finance with the application of other robust methodologies of CS-ARDL in different regions. Moreover, other important variables such as institutional quality, financial development, and economic policy uncertainty can be included in the model.

**Author Contributions:** P.X. conducted the analysis and revised the paper. J.Z. curated the data with theoretical additions and proofread the paper. U.M. wrote the initial version of manuscript. All authors have read and agreed to the published version of the manuscript.

**Funding:** This research received no external funding.

**Institutional Review Board Statement:** Not applicable.

**Informed Consent Statement:** Not applicable.

**Data Availability Statement:** Data will be available on request.

**Conflicts of Interest:** The authors declare no conflict of interest.

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
