# Peer review of "How Do Green Investments, Foreign Direct Investment, and Renewable Energy Impact CO2 Emissions? Measuring the Role of Education in E-7 Nations"

_sustainability, doi:10.3390/su151914052_

Round 1

Reviewer 1 Report

 Title;  Does green finance really green: estimating the roles of education, FDI, renewable energy for CO2 emissions in E-7 nations?”

Major comments

- The abstract is too short and does not provide useful details to the readers.

-The Introduction should highlight the relevance of the topic, the novelty of the results, the importance of policy implications, the sample’s choice, the methodology’s appropriateness, the data used, the contribution to the literature, and the limitations of the study.

- Figure 1. Variables. The authors must explain the purpose of this figure

- The novel contribution of this study is not clear

- The limitation of the study is clear.

-Check the missed references

- Suggest more valuable suggestions and recommendation for the policy makers in E-7 nations.

-Proofreading by a native speaker is required

 Title;  “Does green finance really green: estimating the roles of education, FDI, renewable energy for CO2 emissions in E-7 nations?”

Major comments

- The abstract is too short and does not provide useful details to the readers.

-The Introduction should highlight the relevance of the topic, the novelty of the results, the importance of policy implications, the sample’s choice, the methodology’s appropriateness, the data used, the contribution to the literature, and the limitations of the study.

- Figure 1. Variables. The authors must explain the purpose of this figure

- The novel contribution of this study is not clear

- The limitation of the study is clear.

-Check the missed references

- Suggest more valuable suggestions and recommendation for the policy makers in E-7 nations.

-Proofreading by a native speaker is required

Author Response

Dear reviewer, Thank you for your valuable comments. We incorporated the suggestions and believed that the quality of the manuscript has been improved.  

Major comments

- The abstract is too short and does not provide useful details to the readers.

The abstract has been revised for useful details to the readers.

-The Introduction should highlight the relevance of the topic, the novelty of the results, the importance of policy implications, the sample’s choice, the methodology’s appropriateness, the data used, the contribution to the literature, and the limitations of the study.

 The introduction has been revised carefully to highlight the novelty, sample choice, methodology appropriateness and contribution of the work.

- Figure 1. Variables. The authors must explain the purpose of this figure.

The only purpose of this image to present the conceptual framework by the estimated variables.

- The novel contribution of this study is not clear

The introduction is revised now to add the contribution of the work.

- The limitation of the study is clear.

The limitation of the work has been added.

-Check the missed references

References are checked now.

- Suggest more valuable suggestions and recommendation for the policy makers in E-7 nations.

Policy section has been revised now.

-Proofreading by a native speaker is required

The document is proofread now.

Comments on the Quality of English Language

Reviewer 2 Report

Thank you for submitting your research article. However, upon thorough evaluation, several significant issues have been identified that need to be addressed. Firstly, there is a mismatch between the research content and the article's title, necessitating clarification and alignment. Furthermore, the research contribution lacks practical economic significance and would benefit from incorporating unexplored areas of research along with addressing practical problems. Additionally, it is essential to confirm the research design, including factors such as the research period and the measurement of green finance. Lastly, the article lacks an economic interpretation. Considering these crucial factors, the article is rejected. Thank you for giving us the opportunity to review your work.

In order to improve the paper, we recommend referring to existing research in the field.

Moderate editing of English language required!

Author Response

Dear esteemed reviewer, Thank you for your precious time to read this work. After reading your objections, we revise this work thoroughly. From title, introduction, theoretical framework, methodology, research design, to policy suggestions, we revised it. We hope that now this work will fulfil your requirement.

Thank you for submitting your research article. However, upon thorough evaluation, several significant issues have been identified that need to be addressed. Firstly, there is a mismatch between the research content and the article's title, necessitating clarification and alignment. Furthermore, the research contribution lacks practical economic significance and would benefit from incorporating unexplored areas of research along with addressing practical problems. Additionally, it is essential to confirm the research design, including factors such as the research period and the measurement of green finance. Lastly, the article lacks an economic interpretation. Considering these crucial factors, the article is rejected. Thank you for giving us the opportunity to review your work.

In order to improve the paper, we recommend referring to existing research in the field.

Comments on the Quality of English Language

Moderate editing of English language required!

The manuscript has been proofread for any mistakes.

Reviewer 3 Report

The paper investigates an interesting issue and it is well presented. However, I suggest the author(s) update the literature discussions by adding some recently published works. Please see below a suggested paper:

Löffler, K.U., Petreski, A. & Stephan, A. Drivers of green bond issuance and new evidence on the “greenium”. Eurasian Econ Rev 11(1), 1–24 (2021). https://doi.org/10.1007/s40822-020-00165-y

On the other hand, it would be good if the author(s) compare their results with the results of the previous studies. 

The motivation and contribution of the paper should be written more clearly.

Author Response

Dear esteemed reviewer, thank you for your valuable suggestions.

The paper investigates an interesting issue and it is well presented. However, I suggest the author(s) update the literature discussions by adding some recently published works. Please see below a suggested paper:

Löffler, K.U., Petreski, A. & Stephan, A. Drivers of green bond issuance and new evidence on the “greenium”. Eurasian Econ Rev 11(1), 1–24 (2021). https://doi.org/10.1007/s40822-020-00165-y

We read and cite this suggested work and revised the manuscript as advised.

On the other hand, it would be good if the author(s) compare their results with the results of the previous studies. 

The results are compared with the previous published works.

The motivation and contribution of the paper should be written more clearly.

The motivation and contribution are revised now more clearly.

Reviewer 4 Report

This article examines the nexus between education, the use of renewable energy, green financing, and foreign direct investment and CO2 emissions, in the context of E-7 countries. The research might have marginal potentials for practical implications. However, some aspects must be addressed in order to improve the overall quality of the paper:

1. The title mismatches with the research content. It is suggested to revise the title. The title seems to focus on green finance. In the specific analysis, green finance is not the key topic.

2. The sample period is from 2000 to 2019. The observations in this study are just 140. I think it is not enough for analyzing the relationship between education, GDP, the use of renewable energy, green financing, and foreign direct investment and CO2 emissions.  It is suggested to expanded to 2022 or 2021.

3. The novelties of this manuscript should be more clearly stated in the introduction part. In the present version, I think its contribution is not enough to be published in Sustainability.

4. The theoretic analysis is too weak. The theoretical framework is strange. Why does it entitle Variables (see Fig.1)

5. The empirical analysis is not enough.

6. the policy implications are common sense. Please revise it.

It also lacks of research limitations and future prospects. It is also suggested to add it.

The language needs minor revision.

Author Response

Dear esteemed reviewer, we are thankful for your detailed comments. We revised the manuscript in accordance to your comments. We believe that the quality of this work has been improved and it will fulfil your requirements now. We will be happy to be welcoming further comments if any.

This article examines the nexus between education, the use of renewable energy, green financing, and foreign direct investment and CO2 emissions, in the context of E-7 countries. The research might have marginal potentials for practical implications. However, some aspects must be addressed in order to improve the overall quality of the paper:

  1. The title mismatches with the research content. It is suggested to revise the title. The title seems to focus on green finance. In the specific analysis, green finance is not the key topic.

Dear reviewer, now the title has been revised.

  1. The sample period is from 2000 to 2019. The observations in this study are just 140. I think it is not enough for analyzing the relationship between education, GDP, the use of renewable energy, green financing, and foreign direct investment and CO2 emissions.  It is suggested to expanded to 2022 or 2021.

Dear reviewer, thank you for this important suggestion. Now the data has been increased till 2021 and whole analysis is revised.

  1. The novelties of this manuscript should be more clearly stated in the introduction part. In the present version, I think its contribution is not enough to be published in Sustainability.

The introduction part is revised now to add the novelties.

  1. The theoretic analysis is too weak. The theoretical framework is strange. Why does it entitle Variables (see Fig.1)

This part is revised now and the figure represents the conceptual framework of the study.

  1. The empirical analysis is not enough.

The empirical analysis is enhanced now.

  1. the policy implications are common sense. Please revise it.

Policy implications are revised now.

It also lacks of research limitations and future prospects. It is also suggested to add it.

The limitations of the work are added now.

Comments on the Quality of English Language

The language needs minor revision.

Reviewer 5 Report

This paper investigates the impacts of green finance, the roles of education, FDI, renewable energy for CO2 emissions in E-7 nations.

However, I have some major suggestions to improve the quality of the manuscript.

The introduction is well written, and the motivation of the study is added carefully.

1.     Something is missing in the title please try to use an appropriate word to refine it.

2.     Authors are advised to write overall suggestions for this study in the abstract.

3.     Similarly, the authors must try to discuss the study's contribution briefly.

4.     The authors provide a comprehensive literature review; a paragraph can be added to define the gap.

5.     The result section should be improved with some recent citations.

6.     The conclusion and policy should be revised.

7.     Please check the references in journal format.

8.     I have seen some grammar typos; please read carefully the whole document.

I have seen some grammar typos; please read carefully the whole document.

Author Response

Dear reviewer, thank you for your important suggestions. Your suggestions have been included carefully.

This paper investigates the impacts of green finance, the roles of education, FDI, renewable energy for CO2 emissions in E-7 nations.

However, I have some major suggestions to improve the quality of the manuscript.

The introduction is well written, and the motivation of the study is added carefully.

  1. Something is missing in the title please try to use an appropriate word to refine it.

Title has been revised now.

  1. Authors are advised to write overall suggestions for this study in the abstract.

The abstract is revised now.

  1. Similarly, the authors must try to discuss the study's contribution briefly.

The contribution has been enhanced now.

  1. The authors provide a comprehensive literature review; a paragraph can be added to define the gap.

Few lines have been added to define the gap.

  1. The result section should be improved with some recent citations.

The result section has been revised now.

  1. The conclusion and policy should be revised.

This section is also revised now.

  1. Please check the references in journal format.

These references are also revised now.

  1. I have seen some grammar typos; please read carefully the whole document.

The document has been proofread for mistakes.

Comments on the Quality of English Language

I have seen some grammar typos; please read carefully the whole document.

Reviewer 6 Report

This paper examines the effects of the economy, the use of renewable energy, green financing, foreign direct investment and education on carbon emissions. Different from the conclusions of previous studies, this study finds that green finance is not green. In general, from theory to model, this is an economic research paper with reasonable structure and clear logic. However, there are still several questions that I would like to discuss with the author.

1. First of all, the title of this article, since it discusses the moderating role of education, it should be reflected separately or not.

2. The second question is the most worthy of discussion. The green finance index in this paper is synthesized by principal component analysis, which is described in lines 310 and 311 as follows: “green finance (A composite index by using principal component analysis using the variables of number of patents, energy efficiency, and renewable energy)”. I don't see any authoritative literature or theoretical support for why you do this. I think the following literature will give you deeper thinking.

Dong Qiu, Tingyi Liu. Multi-indicator comprehensive evaluation: reflection on methodology[J]. Data Science in Finance and Economics, 2021, 1(4): 298-312. doi: 10.3934/DSFE.2021016

3. The concept of green finance is more fully discussed in lines 151-171, but I think the following document is still worth reading for the author.

Rupsha Bhattacharyya. Green finance for energy transition, climate action and sustainable development: overview of concepts, applications, implementation and challenges[J]. Green Finance, 2022, 4(1): 1-35. doi: 10.3934/GF.2022001

4. Although the article has done a robustness test, it may not be enough. In view of the conclusions of this paper, it is necessary to replace a measure of green finance with a robustness test.

5. When the paper carries on the theoretical analysis, it adopts the analysis between the two variables. There are also literature discussing the relationship between multiple variables in the article, which is very valuable for reference, but is ignored by the author, such as the literature discussing FDI, financial development and economic growth.

Mustafa Hassan Mohammad Adam. Nexus among foreign direct investment, financial development, and sustainable economic growth: Empirical aspects from Sudan[J]. Quantitative Finance and Economics, 2022, 6(4): 640-657. doi: 10.3934/QFE.2022028

Pointing out these problems does not mean that this is a substandard paper. It is a good paper that can be published if it is properly addressed.

Extensive editing of English language required!

Author Response

Dear esteemed reviewer, we are thankful for your valuable time and important suggestions. We include all your valuable suggestions and believe that the quality of this work has been enhanced.

This paper examines the effects of the economy, the use of renewable energy, green financing, foreign direct investment and education on carbon emissions. Different from the conclusions of previous studies, this study finds that green finance is not green. In general, from theory to model, this is an economic research paper with reasonable structure and clear logic. However, there are still several questions that I would like to discuss with the author.

  1. First of all, the title of this article, since it discusses the moderating role of education, it should be reflected separately or not.

Dear esteemed reviewer, now the title is revised to reflect each variable.

  1. The second question is the most worthy of discussion. The green finance index in this paper is synthesized by principal component analysis, which is described in lines 310 and 311 as follows: “green finance (A composite index by using principal component analysis using the variables of number of patents, energy efficiency, and renewable energy)”. I don't see any authoritative literature or theoretical support for why you do this. I think the following literature will give you deeper thinking.

Dear reviewer, this work follows the recent work of (Musah et al. 2022) to construct the green investment index. Thank you so much for suggesting me the relevant study.

Dong Qiu, Tingyi Liu. Multi-indicator comprehensive evaluation: reflection on methodology[J]. Data Science in Finance and Economics, 2021, 1(4): 298-312. doi: 10.3934/DSFE.2021016

  1. The concept of green finance is more fully discussed in lines 151-171, but I think the following document is still worth reading for the author.

Dear reviewer, now the discussion on green investment is enhanced after reading your suggested article.

Rupsha Bhattacharyya. Green finance for energy transition, climate action and sustainable development: overview of concepts, applications, implementation and challenges[J]. Green Finance, 2022, 4(1): 1-35. doi: 10.3934/GF.2022001

  1. Although the article has done a robustness test, it may not be enough. In view of the conclusions of this paper, it is necessary to replace a measure of green finance with a robustness test.

Dear reviewer, thank you for important suggestion. Now this work applies robustness test with updated data of patents on environmental technologies.

  1. When the paper carries on the theoretical analysis, it adopts the analysis between the two variables. There are also literature discussing the relationship between multiple variables in the article, which is very valuable for reference, but is ignored by the author, such as the literature discussing FDI, financial development and economic growth.

The theoretical analysis has been revised to add the discussion about all analyzed variables.

Mustafa Hassan Mohammad Adam. Nexus among foreign direct investment, financial development, and sustainable economic growth: Empirical aspects from Sudan[J]. Quantitative Finance and Economics, 2022, 6(4): 640-657. doi: 10.3934/QFE.2022028

Pointing out these problems does not mean that this is a substandard paper. It is a good paper that can be published if it is properly addressed.

Thank you for your compliment.

Comments on the Quality of English Language

Extensive editing of English language required!

We proofread this work thoroughly for grammar mistakes.

Round 2

Reviewer 1 Report

Accept in present form

Author Response

Dear reviewer thank you. 

Reviewer 2 Report

,Some redundant references need to be deleted, and the literature review needs to be more adequate:

Rupsha Bhattacharyya. Green finance for energy transition, climate action and sustainable development: overview of concepts, applications, implementation and challenges[J]. Green Finance, 2022, 4(1): 1-35. doi: 10.3934/GF.2022001

 Lu Liu, Ming Liu. How does the digital economy affect industrial eco-efficiency? Empirical evidence from China[J]. Data Science in Finance and Economics, 2022, 2(4): 371-390. doi: 10.3934/DSFE.2022019

Li, Z., Huang, Z., & Su, Y. (2023). New media environment, environmental regulation and corporate green technology innovation: Evidence from China. Energy Economics, 119, 106545. doi:10.1016/j.eneco.2023.106545

Moderate editing of English language required!

Author Response

Reviewer 2

Some redundant references need to be deleted, and the literature review needs to be more adequate:

Rupsha Bhattacharyya. Green finance for energy transition, climate action and sustainable development: overview of concepts, applications, implementation, and challenges[J]. Green Finance, 2022, 4(1): 1-35. doi: 10.3934/GF.2022001

 Lu Liu, Ming Liu. How does the digital economy affect industrial eco-efficiency? Empirical evidence from China[J]. Data Science in Finance and Economics, 2022, 2(4): 371-390. doi: 10.3934/DSFE.2022019

Li, Z., Huang, Z., & Su, Y. (2023). New media environment, environmental regulation and corporate green technology innovation: Evidence from China. Energy Economics, 119, 106545. doi:10.1016/j.eneco.2023.106545

Dear esteemed reviewer, thank you for suggesting these studies. We studied and cite these works in the relevant place. Other redundant references are removed now.

Reviewer 4 Report

congrtas, it is improved a lot.

english language is good

Author Response

Thank you 

Reviewer 5 Report

Thank you! The author has made significant changes to improve the quality of the manuscript. Therefore, the current version of the manuscript is accepted. 

Author Response

Thank you

Reviewer 6 Report

The article is revised according to comments. Additional references to consider:

 Li Z., Chen H. & Mo B. (2022), Can digital finance promote urban innovation? Evidence from China, Borsa Istanbul Review, https://doi.org/10.1016/j.bir.2022. 10.006.

Li, Z., Liao, G., & Albitar, K. (2020). Does corporate environmental responsibility engagement affect firm value? The mediating role of corporate innovation. Business Strategy and the Environment, 29(3), 1045-1055. doi: 10.1002/bse.2416 

Moderate editing of English language required!

Author Response

Reviewer 6

The article is revised according to comments. Additional references to consider:

 Li Z., Chen H. & Mo B. (2022), Can digital finance promote urban innovation? Evidence from China, Borsa Istanbul Review, https://doi.org/10.1016/j.bir.2022. 10.006.

Li, Z., Liao, G., & Albitar, K. (2020). Does corporate environmental responsibility engagement affect firm value? The mediating role of corporate innovation. Business Strategy and the Environment, 29(3), 1045-1055. doi: 10.1002/bse.2416

Dear esteemed reviewer, thank you for suggesting these relevant studies. These studies are cited with their findings.